# Safety of Rigid Bronchoscopy for Therapeutic Intervention at the Intensive Care Unit Bedside

**DOI:** 10.3390/medicina58121762

**Published:** 2022-11-30

**Authors:** Sang Hyuk Kim, Boksoon Chang, Hyun Joo Ahn, Jie Ae Kim, Mikyung Yang, Hojoong Kim, Byeong-Ho Jeong

**Affiliations:** 1Division of Pulmonary, Allergy, and Critical Care Medicine, Department of Internal Medicine, Hallym University Kangnam Sacred Heart Hospital, Hallym University College of Medicine, Seoul 07441, Republic of Korea; 2Department of Pulmonary, Allergy and Critical Care Medicine, Kyung Hee University Hospital at Gangdong, School of Medicine, Kyung Hee University, Seoul 05278, Republic of Korea; 3Department of Anesthesiology and Pain Medicine, Samsung Medical Center, Sungkyunkwan University School of Medicine, Seoul 06351, Republic of Korea; 4Division of Pulmonary and Critical Care Medicine, Department of Medicine, Samsung Medical Center, Sungkyunkwan University School of Medicine, Seoul 06351, Republic of Korea

**Keywords:** rigid bronchoscopy, intensive care unit, complications

## Abstract

*Background and Objective*: Although rigid bronchoscopy is generally performed in the operating room (OR), the intervention is sometimes emergently required at the intensive care unit (ICU) bedside. The aim of this study is to evaluate the safety of rigid bronchoscopy at the ICU bedside. *Materials and Methods*: We retrospectively analyzed medical records of patients who underwent rigid bronchoscopy while in the ICU from January 2014 to December 2020. According to the location of rigid bronchoscopic intervention, patients were classified into the ICU group (*n* = 171, cases emergently performed at the ICU bedside without anesthesiologists) and the OR group (*n* = 165, cases electively performed in the OR with anesthesiologists). The risk of intra- and post-procedural complications in the ICU group was analyzed using multivariable logistic regression, with the OR group as the reference category. *Results*: Of 336 patients, 175 (52.1%) were moribund and survival was not expected without intervention, and 170 (50.6%) received invasive respiratory support before the intervention. The most common reasons for intervention were post-intubation tracheal stenosis (39.3%) and malignant airway obstruction (34.5%). Although the overall rate of intra-procedural complications did not differ between the two groups (86.0% vs. 80.6%, *p* = 0.188), post-procedural complications were more frequent in the ICU group than in the OR group (24.0% vs. 12.1%, *p* = 0.005). Severe complications requiring unexpected invasive management occurred only post-procedurally and were more common in the ICU group (10.5% vs. 4.8%, *p* = 0.052). In the fully adjusted model, the ICU group had increased odds for severe post-procedural complications, but statistical significance was not observed (odds ratio, 2.54; 95% confidence interval, 0.73–8.88; *p* = 0.144). *Conclusions*: Although general anesthesia is generally considered the gold standard for rigid bronchoscopy, our findings indicate that rigid bronchoscopy may be safely performed at the ICU bedside in selective cases of emergency. Moreover, adequate patient selection and close post-procedural monitoring are required to prevent severe complications.

## 1. Introduction

Although the use of rigid bronchoscopy has declined since the introduction of flexible bronchoscopy, it is an indispensable procedure for therapeutic intervention in the field of interventional pulmonology because it has several advantages over flexible bronchoscopies, such as facilitating more complex procedures, better airway control, and greater capability for suction [1,2]. Rigid bronchoscopy is generally performed in the operating room (OR) under general anesthesia to facilitate the procedure and reduce patient discomfort [3]. In the OR, an anesthesiologist monitors the condition of the patient and adequately adjusts the levels of anesthesia and ventilation. However, there is no standard method for ventilation or level of anesthesia for rigid bronchoscopy [4].

In a critically ill patient, it can be harmful to perform rigid bronchoscopy electively in the OR due to delay in the intervention [5]. Consequently, rigid bronchoscopy sometimes has to be completed at the intensive care unit (ICU) bedside. In these emergency situations, the intervention cannot be performed with an anesthesiologist or with devices to assess the degree of anesthesia. If emergent rigid bronchoscopy could be performed safely at the ICU bedside, it would allow improved management of severe patients who require urgent airway intervention.

Previous studies have reported various anesthetic techniques for rigid bronchoscopy [4,6]. However, these studies have focused on effective induction, placing preference on a smooth intervention rather than its safety from the anesthesiologist’s perspective. Some studies have evaluated the safety of rigid bronchoscopy but did not consider the degree of urgency, location of bronchoscopic intervention, or presence of an anesthesiologist [7,8]. Therefore, we aimed to investigate the safety and outcomes of emergent rigid bronchoscopy for therapeutic intervention at the ICU bedside without an anesthesiologist using tertiary hospital-based retrospective data.

## 2. Materials and Methods

### 2.1. Study Population

We conducted a retrospective cohort study by reviewing medical records of patients who underwent rigid bronchoscopy while admitted to the ICU from January 2014 to December 2020 in Samsung Medical Center (a 1960-bed university-affiliated tertiary referral hospital in Seoul, Republic of Korea). During the study period, we had a hospital-wide medical emergency team (MET) for the rapid response system, which consisted of dedicated intensivist physicians, including critical care fellows and attending intensivists [9]. Activation of MET was based on several criteria (respiration, circulation, and neurology), and respiratory criteria include a respiratory rate ≥30/min, oxygen saturation < 90% for 5 min despite previous oxygen administration, acute hypercapnic acidosis, and stridor or use of respiratory accessory muscle. Activation was also allowed when the medical staff was concerned about changes in their patient’s clinical condition, even in the absence of physiological disorders that meet the criteria. Since August 2016, the MET has been activated automatically using a modified early warning score [10]. When the MET decided to transfer the patient to the ICU, the interventional pulmonologists (H. Kim and B.-H. Jeong) carefully decided whether an emergency procedure at the ICU bedside was necessary or an elective procedure in the OR was possible according to the patient’s condition on a case-by-case basis. In general, we made decisions for an emergent procedure based on the following criteria: (a) patients with respiratory failure (PaO_2_/FiO_2_ ratio < 200 mmHg, PaCO_2_ > 50 mmHg, or arterial pH < 7.3) despite invasive mechanical ventilation, (b) not mechanically ventilated patients who were expected of difficulties for tracheal intubation due to tracheal and/or both main bronchial obstruction, and (c) patients with corrosive airway foreign body.

According to the location of rigid bronchoscopic intervention, patients were classified into (a) the ICU group, cases emergently performed at the ICU bedside without anesthesiologists, or (b) the OR group, cases electively performed in the OR with anesthesiologists.

### 2.2. Anesthesia and Airway Intervention Techniques

We used different anesthetic techniques according to the location of intervention as follows. In the OR, the anesthesiologist performed general anesthesia using target-controlled infusion (Orchestra^®^; Fresenius Vial, Brezins, France) with propofol and remifentanil under bispectral index (Aspect Medical Systems, Norwood, MA, USA) monitoring. During the procedure, the anesthesiologist monitored vital signs, and a reversal agent for neuromuscular block agents (NMBA) was administered at the end of the procedure as soon as extubation was possible. At the ICU bedside, anesthesia was induced by the operator with a bolus dose of sedatives, opioids, and NMBA. The assistant monitored vital signs and administered additional drugs depending on the status of the patient. A reversal agent of NMBA was not routinely used in the ICU because the drug was not generally prepared.

Rigid bronchoscopy was performed following standard techniques [11]. After anesthesia, the patient was intubated using a rigid bronchoscope tube (Bryan Co., Woburn, MA, USA, or Karl-Storz, Tuttlingen, Germany). Patients were manually ventilated by an anesthesiologist in the OR and by an assistant at the ICU bedside. Various airway intervention methods were used, such as silicone stent insertion, mechanical debulking, bougienation with a balloon and rigid scope, and laser cauterization, based on the characteristics of the lesion and the reasons for intervention [12].

### 2.3. Adverse Events

We investigated procedure-related complications to assess the safety of rigid bronchoscopy. We divided complications into intra- and post-procedural complications. Intra-procedural complications consisted of hypertension, hypotension, tachycardia, and hypoxia. Hypertension was defined as episodes of systolic blood pressure >160 mmHg [13]. Hypotension was defined as episodes of systolic blood pressure <90 mmHg or administration of vasoactive drugs or fluid to correct hypotension [14,15]. Tachycardia was defined as episodes of heart rate >110 beats per min [16]. Hypoxia was defined as episodes of saturation of oxygen <92% [17]. Post-procedural complications consisted of respiratory failure, atelectasis, pneumonia, newly developed arrhythmia, massive bleeding, pleural effusion, and pneumothorax within three days after the intervention [14]. Respiratory failure was defined as unplanned mechanical ventilation. Pneumonia was defined as the presence of new lung infiltrates on the chest radiograph with fever (body temperature ≥38 °C). Newly developed arrhythmias were defined as a new presentation of atrial fibrillation, bundle branch block, sustained ventricular tachycardia, supraventricular tachycardia, or ventricular fibrillation. Atelectasis, pleural effusion, and pneumothorax were defined using formal readings reported by a chest radiologist. Major bleeding was defined as procedure-related bleeding requiring transfusion [18].

Intra- and post-procedural complications were classified by severity into (a) mild, no additional management; (b) moderate, need for unexpected non-invasive management such as medical therapy; and (c) severe, need for unexpected invasive management.

### 2.4. Data Collection

We retrospectively collected the following data: demographics, patient status before the intervention, disease characteristics, bronchoscopic findings, procedure details, anesthetics used, and intra- and post-procedural findings. Performance status was assessed using the American Society of Anesthesiologists (ASA) physical status classification [19]. In short, ASA class 3 indicates severe systemic disease with substantive functional limitations, class 4 indicates severe systemic disease that is a constant threat to life, and class 5 is moribund, for which survival is not expected without intervention. The severity of stenosis was defined using the Myer–Cotton stenosis grading system: Grade II, 51% to 70% luminal stenosis; Grade III, 71% to 99% luminal stenosis; and Grade IV, no lumen [20]. Patients who underwent intervention for fistula (*n* = 7) or stent removal without airway stenosis (*n* = 1) were classified as Grade II.

### 2.5. Statistical Analysis

Data are expressed as numbers (%) for categorical variables and median (interquartile range, IQR) for continuous variables. Differences between the two groups were analyzed using Pearson’s chi-square test or Fisher’s exact test for categorical variables and the Mann–Whitney test for continuous variables.

Multivariable logistic regression analysis was used to adjust for potential confounding factors in intra- and post-procedural complications according to the location of bronchoscopic intervention, with the OR group as the reference category. Four models were constructed: Model 1 was adjusted for selected variables with *p* < 0.20 in the comparison between the two groups; Model 2 was adjusted for demographics; Model 3 was adjusted for variables that were generally associated with disease severity; Model 4 was adjusted for all of the abovementioned variables. In this analysis, the PaO_2_/FiO_2_ ratio was input after being converted into the following categorical variables: (a) <200 mmHg, (b) 200–299 mmHg, and (c) ≥300 mmHg. Interventions under extracorporeal membrane oxygenation (ECMO) support were classified as <200 mmHg, and interventions without arterial blood gas analysis were classified as ≥300 mmHg. In the reason for intervention category, airway foreign body, relapsing polychondritis, and post-tuberculous tracheobronchial stenosis were analyzed together because there were too few cases of each to analyze properly in multivariable analysis. For similar reasons, the site of the lesion was included in the multivariate analysis only as a single versus extended lesion. The time interval from the end of the intervention to the first extubation was converted into (a) ≤15 min, (b) 16–60 min, (c) ≥61 min [21] with patients who had no plans for extubation and would remain on a home ventilator or were transferred to another hospital immediately after the procedure classified as ≥61 min.

All tests were two-sided, and a *p* < 0.05 was considered significant. All statistical analyses were performed using SPSS (IBM SPSS Statistics ver. 27, Chicago, IL, USA).

## 3. Results

During the study period, 2707 cases of rigid bronchoscopy were performed (Figure 1). In a total of 2707 cases, 336 underwent rigid bronchoscopy after admittance to the ICU, and 165 (49.1%) and 171 (50.9%) were classified into the OR and ICU groups, respectively.

### 3.1. Baseline Characteristics

The baseline characteristics of the study population are shown in Table 1. The median age was 63 years, and 158 (47.0%) patients were male. Body mass index was higher in the ICU group than in the OR group (22.0 vs. 20.8 kg/m^2^, *p* = 0.025). The most common comorbidity was cancer (42.0%), followed by diabetes mellitus (28.9%), cerebrovascular disease (17.0%), and congestive heart failure (15.8%). Half of the patients (52.1%) were ASA V, but there was no statistical difference in the ASA performance status between the two groups. The PaO_2_/FiO_2_ ratio was lower (286 vs. 370 mmHg, *p* < 0.001), and the proportion of ECMO before intervention (7.6% vs. 1.2%, *p* = 0.005) was higher in the ICU group than in the OR group. The most common reason for intervention was post-intubation or tracheostomy tracheal stenosis (39.3%), followed by malignant central airway obstruction (34.5%) and postoperative tracheobronchial stenosis (7.1%).

### 3.2. Details of Rigid Bronchoscopy and Anesthetic Drugs

The ICU group had a shorter time interval between ICU admission and bronchoscopic intervention (18 vs. 41 h, *p* < 0.001) and a longer intervention duration (20 vs. 16 min, *p* = 0.008) than the OR group (Table 2). The proportion of single lesions among all patients was 74.7%, and the trachea was the most common disease site (79.8%) with single lesions (61.0%) and extended lesions (18.8%) combined. The site of the lesion did not differ between the two groups (*p* = 0.234), but stenosis was more severe in the ICU group than in the OR group (29.8% vs. 18.2% for Myer and Cotton grade IV, *p* = 0.014). The most common procedure was stent insertion (65.8%), followed by stent change or repositioning (22.3%) and tumor removal (17.0%).

There were significant differences between the two groups in the drugs used for anesthesia. In the OR group, almost all patients were induced with propofol (98.2%) and remifentanil (97.0%), and some were supplemented with midazolam (38.8%). In the ICU group, however, ketamine (55.0%) and midazolam (45.6%) were used as sedatives and fentanyl (85.4%) as the opioid. As an NMBA, rocuronium was the most used agent in both groups (95.2% in the OR group and 68.4% in the ICU group). Reversal agents were used more frequently in the OR group and were rarely used in the ICU group (78.2% vs. 1.2%, *p* < 0.001). Finally, the time interval from the end of intervention to the first extubation was longer in the ICU group than in the OR group (18 vs. 168 min, *p* < 0.001).

### 3.3. Procedure-Related Complications and Other Outcomes

There was no statistically significant difference in the incidence of overall intra-procedural complications between the ICU and OR groups (86.0% vs. 80.6%, *p* = 0.241, Table 3). However, there were significant differences in the details. Hypertension (60.2% vs. 28.5%, *p* < 0.001), tachycardia (60.8% vs. 36.4%, *p* < 0.001), and hypoxia (18.1% vs. 3.6%, *p* < 0.001) were more commonly observed in the ICU group than in the OR group, and the opposite was observed with hypotension (11.1% vs. 53.3%, *p* < 0.001). The overall post-procedural complication rate was significantly higher in the ICU group than in the OR group (24.0% vs. 12.1%, *p* = 0.007). However, there were no statistically significant differences between the two groups in each post-procedural complication.

The severity of intra- and post-procedural complications is visualized in Figure 2. There were no severe intra-procedural complications. However, in a total of 336 patients, severe post-procedural complications occurred in 26 (7.7%), and the rate tended to be higher in the ICU group than in the OR group (10.5% [18/171] vs. 4.8% [8/165], *p* = 0.052). In the total patients, the median length of ICU stay after the intervention was four days, and additional intervention was needed in 18.2%; neither measure showed a statistical difference between the two groups. Despite an ICU mortality of 16 cases (4.8%), there was no procedure-related mortality. The ICU mortality rate did not statistically differ between the two groups (5.8% in the ICU group vs. 3.6% in the OR group, *p* = 0.341).

Comparing the baseline characteristics and procedure details according to presence or absence of severe post-procedural complications (Appendix A), the intervention duration was longer (30 vs. 16 min, *p* < 0.001), and stent change was more common (46.2% vs. 20.3%, *p* = 0.002) in patients with severe post-procedural complications. Finally, patients with severe post-procedural complications showed longer ICU stay (six vs. four days, *p* < 0.001) and more frequent additional intervention (50.0% vs. 15.5%, *p* < 0.001) and had a trend of higher ICU mortality (11.5% vs. 4.2%, *p* = 0.117) than those without.

### 3.4. Risk of Procedure-Related Complications

Table 4 shows whether the ICU group had an increased risk of procedure-related complications. In the crude model, the risk of intra-procedural complications was not statistically higher in the ICU group than in the OR group (unadjusted odds ratio, 1.47; 95% confidence interval [CI], 0.83–2.63; *p* = 0.189). Similar results were observed in various adjusted models (*p* > 0.05 for all). On the other hand, the ICU group had a significantly increased risk of post-procedural complications (unadjusted odds ratio, 2.29; 95% CI, 1.27–4.10; *p* = 0.006) compared to the OR group. This statistical significance was maintained for all models; in a fully adjusted model, the ICU group had a 3.19 adjusted odds ratio (95% CI, 1.43–7.11; *p* = 0.005) with the OR group as a reference.

In cases of severe complications, none were intra-procedural. Increased odds of severe post-procedural complications were observed in the ICU group (unadjusted odds ratio, 2.31; 95% CI, 0.98–5.47; *p* = 0.057). Only the model adjusted for demographic data showed significant results. Finally, the fully adjusted model showed no statistical significance (adjusted odds ratio, 2.54; 95% CI, 0.73–8.88, *p* = 0.144).

## 4. Discussion

Although the ICU group had a lower PaO_2_/FiO_2_ ratio, a higher proportion of ECMO support, more grade IV stenosis, and a longer intervention duration than the OR group, the overall intra-procedural complication rate was similar between the OR and ICU groups. There was only a difference in the incidence of each intra-procedural complication between the two groups. Of note, there was no severe intra-procedural complication, regardless of the location of the bronchoscopic intervention. Post-procedural complications, including those that were severe, occurred more frequently in the ICU group than in the OR group. However, the odds of severe post-procedural complications for the ICU group were not statistically significant after adjusting for all possible confounding factors.

The rigid bronchoscope was a unique diagnostic and therapeutic tool that allowed direct observation of the inside of the airway until the flexible bronchoscope was introduced in the late 1960s [2]. Flexible bronchoscopy is less invasive, does not require general anesthesia, and is easier to observe the peripheral airway than rigid bronchoscopy, so rigid bronchoscope was virtually abandoned [22]. However, Dr Dumon pioneered modern rigid bronchoscopy with the development of silicone stent insertion to treat central airway obstruction in the late 1980s [22,23]. Because rigid bronchoscopy has several advantages over flexible bronchoscopies, such as facilitating mechanical debulking and silicone stent insertion, better airway control, and greater capability for suction, rigid bronchoscopy is irreplaceable in the field of interventional pulmonology [2]. In particular, silicone stents are generally believed to be less injurious to the airway and easier to modify and remove the stent itself than metallic stents [24]. Therefore, rigid bronchoscopy is still used despite the disadvantages of having difficulties in use and requiring general anesthesia. However, there is a limitation to applying the rigid bronchoscopy as an emergency procedure because general anesthesia is required with an anesthesiologist in the operating room, so we analyzed the safety of emergent rigid bronchoscopy at the ICU bedside without an anesthesiologist.

Several studies have reported complications associated with rigid bronchoscopy [8]. However, there is no standardized tool to assess the complications of rigid bronchoscopy, and the rate of complications varies [25,26,27,28]. The subjects of this study were in acute states, such as being ASA class 5 in 52.1% and receiving invasive respiratory supports in 50.6% of total 336 patients, while being in chronic conditions, such as having a median 21.1 kg/m^2^ of body mass index and many comorbidities. In particular, a higher ASA score before the intervention is a risk factor for procedure-related complications [28]. A recent study reported that the rates of intra-procedural hypoxemia and hypotension were 67% and 77% in patients with poor performance status, respectively [7]. In our report, the incidences of intra-procedural hypoxemia and hypotension in patients who underwent rigid bronchoscopy at the ICU bedside were lower at 18% and 11%, respectively. Instead, the incidence of hypertension and tachycardia was high in our study, and the overall incidence of intra-procedural complications was 86%. Although the incidences of these intra-procedural complications were high in our study, they were usually transient and rarely led to serious outcomes. Considering that ICU patients are not easy to transport and may require prompt intervention in an emergency situation, rigid bronchoscopy for therapeutic intervention at the ICU bedside can be a reasonable option for critically ill patients.

Although the overall incidence of intra-procedural complications was similar between the two groups, the characteristics were markedly different. These differences may be due to the differences in anesthetic agents and reversal agents used in the two groups. The occurrence of hypertension, hypotension, and tachycardia was correlated with the known side effects of the drugs mainly used in the two groups [29]. Significantly, more caution should be observed when using ketamine in patients with underlying heart failure, significant arrhythmias, and ischemic cardiomyopathy. Unlike in those complications, the occurrence of intra-procedural hypoxia may be associated with the oxygenation status of the patient before intervention. Our findings showed that the PaO_2_/FiO_2_ ratio was lower in the ICU group than in the OR group, and this may have resulted in more frequent intra-procedural hypoxia in the ICU group. Importantly, although the ICU group had worse conditions than the OR group, all intra-procedural complications were transient and did not require an escalation in management.

The risk of post-procedural complications was higher in the ICU group than in the OR group, possibly due to the use of reversal agents. Delayed extubation after surgery increases the risk of post-procedural complications in various types of surgery [21]. Reversal agents were rarely used in the ICU group, which resulted in delayed extubation. Therefore, it could be postulated that the use of reversal agents may affect the development of post-procedural complications. However, since our study enrolled only a few patients, further studies are needed to prove our hypothesis.

Severe post-procedural complications were associated with poor clinical outcomes, such as longer length of ICU stay and the need for additional interventions (Appendix A), whereas mild to moderate complications did not show a statistical correlation with these clinical outcomes (data not shown). So, we investigated the factors related to the occurrence of severe complications that require unexpected invasive management. Eventually, we noted that stent change or repositioning and longer intervention duration may be associated with severe complications (Appendix A). Stent change or repositioning is usually required when complications occur, such as stent migration or stenosis at both tips of the stent. The procedure is generally performed by widening the stenosis site, removing the granulation tissue and the existing stent, and inserting a new stent. Therefore, a stent change generally takes longer than other procedures. In addition, prolonged intervention duration could be associated with physiological changes that predispose one to respiratory failure [21]. Therefore, careful post-procedural monitoring and management are necessary when the procedure is complicated and takes a long time.

To the best of our knowledge, this is the first study investigating the safety of rigid bronchoscopy at the ICU bedside among critically ill patients. Our findings could serve as a reference for the management of critically ill patients requiring immediate rigid bronchoscopic intervention. However, some limitations should be considered. First, our data were from a single center, where two skilled interventional pulmonologists performed approximately 2700 rigid bronchoscopy cases during the study period. This can limit generalization. Second, differences between the two groups could not be fully adjusted due to the small number of cases. Third, a causal relationship between intervention and the occurrence of complications cannot be asserted due to the retrospective design. However, we meticulously defined the complications to overcome this limitation. Finally, the median time elapsed from ICU admission to the intervention was 18 h in the ICU group, which is hard to see as an emergency situation. Considering the limitation that it was difficult to collect the time points when the operators were contacted retrospectively, we had no choice but to collect data on ICU admission time. However, since 49.1% of the ICU group received the procedure on the weekends or nighttime, and only 13.5% received the procedure during the weekdays in the elective time, we can assume that we performed an emergency procedure with a quick decision in the ICU group.

## 5. Conclusions

In conclusion, there was no difference in the incidence of intra-procedural complications between the two groups, and none of these complications were severe cases requiring additional invasive procedures. Although general anesthesia is generally considered the gold standard for rigid bronchoscopy, rigid bronchoscopy may be safely performed at the ICU bedside in selective cases of emergency. Severe post-procedural complications tended to be higher in the ICU group than in the OR group; however, we think this is within acceptable limits given the need for emergent rigid bronchoscopic intervention. Moreover, adequate patient selection and close post-procedural monitoring are required for successful rigid bronchoscopy at the ICU bedside to prevent severe complications.

## Figures and Tables

**Figure 1 medicina-58-01762-f001:**
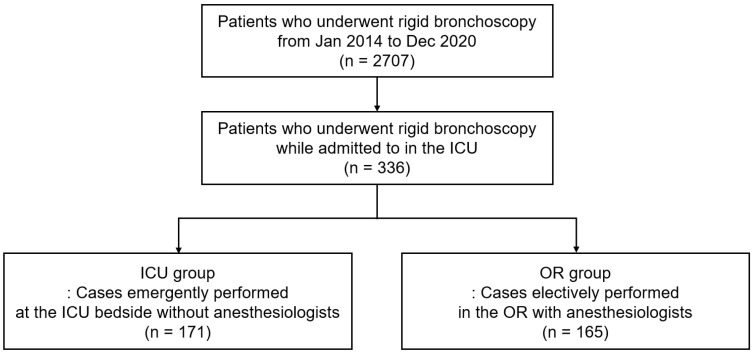
Flow chart of the study. ICU = intensive care unit, OR = operating room.

**Figure 2 medicina-58-01762-f002:**
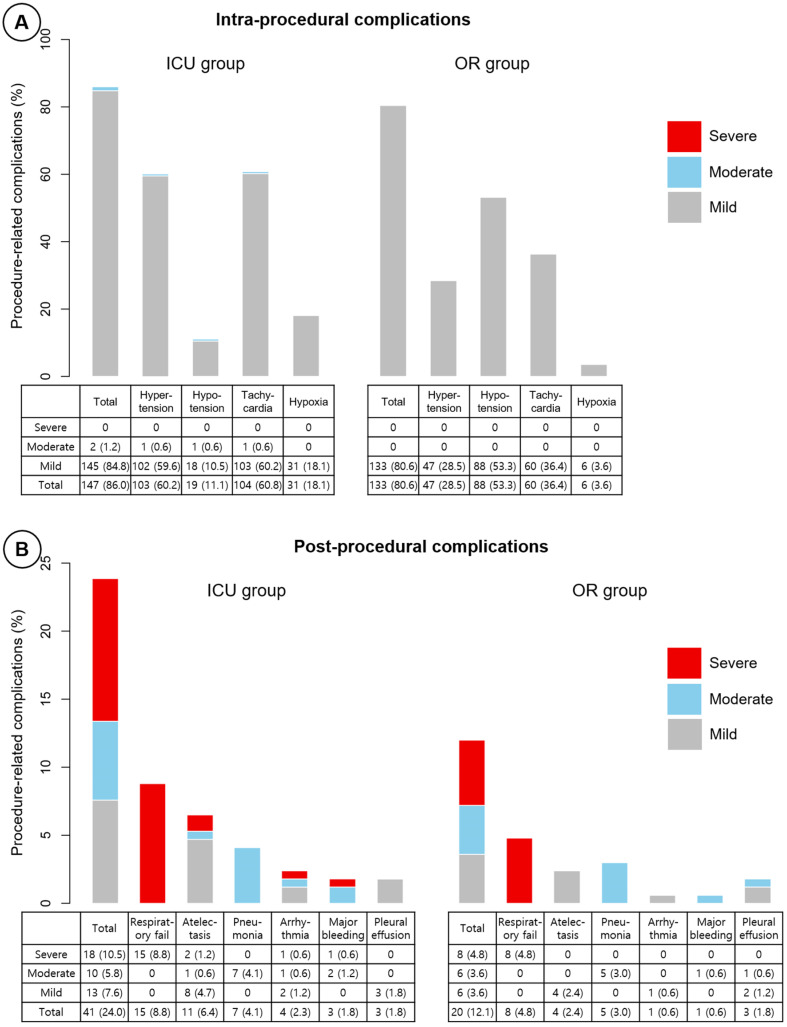
The incidence of (**A**) intra- and (**B**) post-procedural complications according to the location of rigid bronchoscopic intervention. ICU = intensive care unit, OR = operating room.

**Table 1 medicina-58-01762-t001:** Baseline characteristics.

Variables	Total (*n* = 336)	ICU Group (*n* = 171)	OR Group (*n* = 165)	*p*
Age, years	63 (50–74)	63 (50–74)	63 (50–74)	0.651
Male	158 (47.0)	87 (50.9)	71 (43.0)	0.150
BMI, kg/m^2^	21.1 (18.6–24.1)	22.0 (19.3–24.3)	20.8 (18.1–23.9)	0.025
Current or past smoker	138 (41.1)	77 (45.0)	61 (37.0)	0.133
Comorbidities				
Cancer	141 (42.0)	69 (40.4)	72 (43.6)	0.542
Diabetes mellitus	97 (28.9)	48 (28.1)	49 (29.7)	0.742
Cerebrovascular disease	57 (17.0)	26 (15.2)	31 (18.8)	0.382
Congestive heart failure	53 (15.8)	29 (17.0)	24 (14.5)	0.544
Chronic pulmonary disease	28 (8.3)	17 (9.9)	11 (6.7)	0.278
Chronic liver disease	18 (5.4)	9 (5.3)	9 (5.5)	0.938
Performance status ^a^				0.714
ASA III	42 (12.5)	20 (11.7)	22 (13.3)	
ASA IV	119 (35.4)	64 (37.4)	55 (33.3)	
ASA V	175 (52.1)	87 (50.9)	88 (53.3)	
Tracheostomy before the intervention	50 (14.9)	23 (13.5)	27 (16.4)	0.453
Invasive respiratory support before intervention	170 (50.6)	84 (49.1)	86 (52.1)	0.583
Mechanical ventilation without ECMO	155 (46.1)	71 (41.5)	84 (50.9)	0.084
With ECMO	15 (4.5)	13 (7.6)	2 (1.2)	0.005
Arterial blood gas analysis ^b^				
PaO_2_/FiO_2_ ratio, mmHg	332 (241–445)	287 (206–391)	370 (272–463)	<0.001
<200 and/or ECMO	66 (19.6)	48 (28.1)	18 (10.9)	
200–299	78 (23.2)	41 (24.0)	37 (22.4)	<0.001 ^c^
≥300 and/or no results	192 (57.1)	82 (48.0)	110 (66.7)	
PaCO_2_, mmHg	38.5 (33.2–43.9)	38.0 (32.7–44.5)	38.9 (34.1–43.6)	0.684
HCO_3_, mEq/L	25.0 (22.3–28.1)	24.6 (22.1–27.5)	25.4 (22.8–28.4)	0.119
Reason for intervention				
PITS	132 (39.3)	59 (34.5)	73 (44.2)	0.068
MCAO	116 (34.5)	59 (34.5)	57 (34.5)	0.993
POTS	24 (7.1)	12 (7.0)	12 (7.3)	0.928
Airway FB	20 (6.0)	19 (11.1)	1 (0.6)	<0.001
Relapsing polychondritis	17 (5.1)	8 (4.7)	9 (5.5)	0.746
PTBS	14 (4.2)	9 (5.3)	5 (3.0)	0.306
Others ^d^	13 (3.9)	5 (2.9)	8 (4.8)	0.360

Data are expressed as medians (interquartile ranges) for continuous variables and numbers (percentages) for categorical variables. ^a^ ASA III indicates severe systemic disease with substantive functional limitations, ASA IV indicates severe systemic disease that is a constant threat to life, and ASA V is moribund, for which survival is not expected without intervention. ^b^ Excluding 15 patients receiving ECMO support before the intervention and 18 patients without results of arterial blood gas. However, when classifying the PaO_2_/FiO_2_ ratio as a categorical variable, patients with ECMO support were categorized as <200, and patients without results were categorized as ≥300. ^c^ This *p*-value is for categorized PaO_2_/FiO_2_ ratios. ^d^ Persistent air leak and/or fistula (*n* = 7), benign tumors (*n* = 2), inhalation injury (*n* = 2), and tracheomalacia due to mucopolysaccharidosis (*n* = 2). ICU = intensive care unit, OR = operating room, BMI = body mass index, ASA = American Society of Anesthesiologists, ECMO = extracorporeal membrane oxygenation, PaO_2_ = arterial partial pressure of oxygen, FiO_2_ = fraction of inspired oxygen, PaCO_2_ = arterial partial pressure of carbon dioxide, PITS = post-intubation or tracheostomy tracheal stenosis, MCAO = malignant central airway obstruction, POTS = postoperative tracheobronchial stenosis, FB = foreign body, PTBS = post-tuberculous tracheobronchial stenosis.

**Table 2 medicina-58-01762-t002:** Details of rigid bronchoscopy and anesthetic agents.

Variables	Total (*n* = 336)	ICU Group (*n* = 171)	OR Group (*n* = 165)	*p*
Time interval from ICU admission to intervention, hours	24 (14–60)	18 (3–36)	41 (19–70)	<0.001
<3 h	41 (12.2)	39 (22.8)	2 (1.2)	<0.001
<6 h	62 (18.5)	59 (34.5)	3 (1.8)	<0.001
Time of intervention				
Weekends or nighttime ^a^	86 (25.6)	84 (49.1)	2 (1.2)	<0.001
Weekdays outside of the elective time ^b^	66 (19.6)	64 (37.4)	2 (1.2)	<0.001
Weekdays in the elective time ^b^	184 (54.8)	23 (13.5)	161 (97.6)	<0.001
Site of lesion				
Single lesion	251 (74.7)	123 (71.9)	128 (77.6)	0.234 ^c^
Trachea	205 (61.0)	97 (56.7)	108 (65.5)	0.101
RMB and/or RBI	23 (6.8)	10 (5.8)	13 (7.9)	0.461
LMB	17 (5.1)	12 (7.0)	5 (3.0)	0.095
Lobar bronchus	6 (1.8)	4 (2.3)	2 (1.2)	0.685
Extended lesion	85 (25.3)	48 (28.1)	37 (22.4)	0.234 ^c^
Trachea and any bronchi	63 (18.8)	37 (21.6)	26 (15.8)	0.167
Both main bronchi	20 (6.0)	11 (6.4)	9 (5.5)	0.705
One main bronchus and contralateral lobar bronchus	2 (0.6)	0	2 (1.2)	0.240
Severity of stenosis ^d^				0.014
II	117 (34.8)	49 (28.7)	68 (41.2)	
III	138 (41.1)	71 (41.5)	67 (40.6)	
IV	81 (24.1)	51 (29.8)	30 (18.2)	
Procedure details				
Stent insertion	221 (65.8)	108 (63.2)	113 (68.5)	0.304
Stent change or reposition	75 (22.3)	30 (17.5)	45 (27.3)	0.032
Tumor removal	57 (17.0)	30 (17.5)	27 (16.4)	0.773
Stent removal	32 (9.5)	19 (11.1)	13 (7.9)	0.313
Bougienation only ^e^	28 (8.3)	9 (5.3)	19 (11.5)	0.038
Tracheostomy	25 (7.4)	12 (7.0)	13 (7.9)	0.764
Foreign body removal	20 (6.0)	19 (11.1)	1 (0.6)	<0.001
Laser cauterization	12 (3.6)	0	12 (7.3)	<0.001
EBV insertion	3 (0.9)	3 (1.8)	0	0.248
Anesthetic agents				
Sedative				
Midazolam	142 (42.3)	78 (45.6)	64 (38.8)	0.205
Ketamine	95 (28.3)	94 (55.0)	1 (0.6)	<0.001
Propofol	181 (53.9)	19 (11.1)	162 (98.2)	<0.001
Opioid				
Fentanyl	149 (44.3)	146 (85.4)	3 (1.8)	<0.001
Remifentanil	164 (48.8)	4 (2.3)	160 (97.0)	<0.001
NMBA				
Vecuronium	23 (6.8)	22 (12.9)	1 (0.6)	<0.001
Cisatracurim	26 (7.7)	25 (14.6)	1 (0.6)	<0.001
Succinylcholine	8 (2.4)	3 (1.8)	5 (3.0)	0.443
Rocuronium	275 (81.8)	118 (69.0)	157 (95.2)	<0.001
Reversal agents for NMBA	131 (39.0)	2 (1.2)	129 (78.2)	<0.001
Sugammadex	119 (35.4)	2 (1.2)	117 (70.9)	<0.001
Neostigmine and glycopyrrolate	12 (3.6)	0 (0)	12 (7.3)	<0.001
Intervention duration, minutes	17 (12–27)	20 (12–30)	16 (12–22)	0.008
Time interval from the end of intervention to the first extubation, minutes ^f^	85 (15–349)	168 (62–1002)	18 (8–124)	<0.001
≤15 min	80 (23.8)	7 (4.1)	73 (44.2)	
16–60 min	63 (18.8)	32 (18.7)	31 (18.8)	<0.001 ^g^
≥61 min	193 (57.4)	132 (77.2)	61 (37.0)	

Data are expressed as medians (interquartile ranges) for continuous variables and numbers (percentages) for categorical variables. ^a^ The definition of night is 6:00 p.m. to 7:00 a.m. ^b^ During the study period, we had a regular slot for the procedure on Tuesdays and Thursdays from 7:00 a.m. to 1:00 p.m. So, the definition of the elective time is 7:00 a.m. to 1:00 p.m. on Tuesdays and Thursdays. ^c^ This *p*-value is for the site of a lesion classified as single or extended lesion. ^d^ Myer and Cotton grade. There was no stenosis in four patients in each of the two groups (persistent air leak and/or fistula = 7, intolerance to tracheal stent due to bronchospasm and mucostasis = 1). They were classified as Grade II. ^e^ Bougienation involves balloon dilatation and/or mechanical bougie using a rigid bronchoscope without stent insertion or tumor removal. ^f^ There were nine cases with home ventilators (ICU group = 5, OR group = 4) and seven cases that were transferred to other hospitals with mechanical ventilators immediately after the intervention (ICU group = 5, OR group = 2). These cases were classified as ≥61 min. ^g^ This *p*-value is for the categorized variable. ICU = intensive care unit, OR = operating room, RMB = right main bronchus, RBI = right bronchus intermedius, LMB = left main bronchus, EBV = endobronchial valve, NMBA = neuromuscular blocking agent.

**Table 3 medicina-58-01762-t003:** Clinical outcomes.

Variables	Total (*n* = 336)	ICU Group (*n* = 171)	OR Group (*n* = 165)	*p*
Intra-procedural complication	280 (83.3)	147 (86.0)	133 (80.6)	0.188
Hypertension	150 (44.6)	103 (60.2)	47 (28.5)	<0.001
Hypotension	107 (31.8)	19 (11.1)	88 (53.3)	<0.001
Tachycardia	164 (48.8)	104 (60.8)	60 (36.4)	<0.001
Hypoxia	37 (11.0)	31 (18.1)	6 (3.6)	<0.001
Severe intra-procedural complication	0 (0.0)	0 (0.0)	0 (0.0)	-
Post-procedural complication	61 (18.2)	41 (24.0)	20 (12.1)	0.005
Respiratory failure	23 (6.8)	15 (8.8)	8 (4.8)	0.155
Atelectasis	15 (4.5)	11 (6.4)	4 (2.4)	0.075
Pneumonia	12 (3.6)	7 (4.1)	5 (3.0)	0.600
Pleural effusion	6 (1.8)	3 (1.8)	3 (1.8)	1.000
Newly developed arrhythmia	5 (1.5)	4 (2.3)	1 (0.6)	0.372
Major bleeding	4 (1.2)	3 (1.8)	1 (0.6)	0.623
Pneumothorax	0 (0.0)	0 (0.0)	0 (0.0)	-
Severe post-procedural complication	26 (7.7)	18 (10.5)	8 (4.8)	0.052
Severe respiratory failure	23 (6.8)	15 (8.8)	8 (4.8)	0.155
Severe atelectasis	2 (0.6)	2 (1.2)	0 (0.0)	0.499
Severe pneumonia	0 (0.0)	0 (0.0)	0 (0.0)	-
Severe pleural effusion	0 (0.0)	0 (0.0)	0 (0.0)	-
Severe newly developed arrhythmia	1 (0.3)	1 (0.6)	0 (0.0)	1.000
Severe major bleeding	1 (0.3)	1 (0.6)	0 (0.0)	1.000
Severe pneumothorax	0 (0.0)	0 (0.0)	0 (0.0)	-
Lengths of ICU stay after intervention, day	4 (3–6)	3 (2–6)	4 (3–6)	0.147
Need for additional intervention	61 (18.2)	30 (17.5)	31 (18.8)	0.767
Mortality				
Procedure-related mortality	0 (0.0)	0 (0.0)	0 (0.0)	-
ICU mortality	16 (4.8)	10 (5.8)	6 (3.6)	0.341

Data are expressed as medians (interquartile ranges) for continuous variables and numbers (percentages) for categorical variables. ICU = intensive care unit, OR = operating room.

**Table 4 medicina-58-01762-t004:** The risk of procedure-related complications for emergent rigid bronchoscopy at the intensive care unit bedside with the OR group as the reference category.

Model	Intra-Procedural Complications		Post-Procedural Complications		Severe Post-Procedural Complications	
	Odds Ratio (95% CI)	*p*	Odds Ratio (95% CI)	*p*	Odds Ratio (95% CI)	*p*
Crude	1.47 (0.83–2.63)	0.189	2.29 (1.27–4.10)	0.006	2.31 (0.98–5.47)	0.057
Model 1	1.49 (0.71–3.13)	0.295	2.82 (1.30–6.09)	0.009	2.51 (0.75–8.40)	0.136
Model 2	1.72 (0.92–3.22)	0.092	2.50 (1.37–4.55)	0.003	2.68 (1.10–6.47)	0.028
Model 3	1.27 (0.69–2.34)	0.434	2.34 (1.26–4.34)	0.007	2.10 (0.86–5.13)	0.105
Model 4	1.44 (0.66–3.14)	0.359	3.19 (1.43–7.11)	0.005	2.54 (0.73–8.88)	0.144

Model 1 was adjusted for selected variables with *p* < 0.20 in the comparison between two groups (sex, BMI, smoking status, PaO_2_/FiO_2_ ratio, reason for intervention, time interval from ICU admission to intervention, severity of stenosis, intervention duration, and time interval from the end of intervention to the first extubation). Model 2 was adjusted for demographics (age, sex, BMI, smoking status, and reason for intervention). Model 3 was adjusted for variables generally considered for disease severity (performance status, invasive respiratory support, PaO_2_/FiO_2_ ratio, time interval from ICU admission to intervention, site of lesion, and severity of stenosis). Model 4 was adjusted for all of the abovementioned variables (age, sex, BMI, smoking status, performance status, invasive respiratory support, PaO_2_/FiO_2_ ratio, reason for intervention, time interval from ICU admission to intervention, site of lesion, severity of stenosis, intervention duration, and time interval from the end of intervention to the first extubation). OR = operating room, CI = confidence interval, BMI = body mass index, PaO_2_ = arterial partial pressure of oxygen, FiO_2_ = fraction of inspired oxygen, ICU = intensive care unit.

## Data Availability

Data are fully available on reasonable request.

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
