# Peer review of "Safety of Rigid Bronchoscopy for Therapeutic Intervention at the Intensive Care Unit Bedside"

_medicina, 2022, doi:10.3390/medicina58121762_

Round 1

Reviewer 1 Report

1   1. Your populations are almost chronic condition in this article. Why need to perform emergent rigid bronchoscopy? What are the indications of emergent rigid bronchoscopy for therapeutic intervention at the ICU bedside?

2.     2. In view of patient safety, why do you need to perform rigid bronchoscope, especially In ICU bedside.

3.    3. There are many articles of flexible bronchoscopy in therapy with self-expandable metallic stents for tracheobronchial stenosis patients. What is different between rigid bronchoscopy and flexible bronchoscopy, especially the different of complications? What is the benefit of rigid bronchoscopy? Please discuss this issue and cite references

Author Response

Thank you for the insightful comments. We tried our best to respond to your comments one by one. Please see attached file.

Reviewer 2 Report

Dear authors, I read your manuscript with interest. I congratulate you on the quality of the study and the results obtained.

The management of patients with acute airway disease is complicated. Rapid intervention is crucial for the excellent outcome and survival of the patient.

Although I was impressed by the thorough analysis of the data collected, I have some concerns about the interpretation reported in the discussion and conclusions.

Don't you think it may be premature to call bed-side rigid bronchoscopy interventions safe? I agree there are situations where the intervention cannot be procrastinated. Still, looking at the data, I read that in the ICU group, the average time elapsed between admission and intervention was 18 hours [3-36]. It could create a misunderstanding between emergency and urgent, being more than good timing for assessment, transport, and OR preparation. 

I would ask you to more thoroughly justify the choices of procedures performed bed-side because of this finding.

Accordingly, I would remain more cautious in my conclusions, keeping in mind that the gold standard of rigid bronchoscopy continues to be GA with neuromuscular blockade.

Finally, I would ask for your choice of a more readable graph type for Figure 2 and an attempt at fluidization of English.

Congratulations on the excellent work.

Translated with www.DeepL.com/Translator (free version)

Author Response

(The authors gave the same response as above.)

Round 2

Reviewer 1 Report

No any comments